# Combined Toxicity of Glyphosate (Faena®) and Copper to the American Cladoceran *Daphnia exilis*—A Two-Generation Analysis

Miriam Hernández-Zamora, Alma Rodríguez-Miguel, Laura Martínez-Jerónimo and Fernando Martínez-Jerónimo *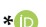

Instituto Politécnico Nacional, Escuela Nacional de Ciencias Biológicas, Lab. de Hidrobiología Experimental, Prol. de Carpio y Plan de Ayala s/n, Col. Santo Tomás, Mexico City 11340, Mexico
* Correspondence: fjeroni@ipn.mx

**Abstract:** Glyphosate and copper are common pollutants in water and soil. Glyphosate is the most used herbicide worldwide; despite being claimed to be a low-toxicity pesticide and easily degraded, several studies have demonstrated glyphosate's occurrence and toxicity in freshwater ecosystems. Copper is an essential micronutrient; however, at high concentrations, it becomes toxic, and it is a persistent contaminant discharged from agricultural and industrial activities. Both glyphosate and Cu are agrochemicals that can occur in aquatic environments and produce stress in aquatic biota. Cladocerans are important zooplankters, and their sensitivity to chemical stressors has been widely documented. In this study, the North American cladoceran *Daphnia exilis* was exposed to glyphosate (Faena®) and Cu mixtures. The effects were assessed in two generations to determine toxicity impairment in the parental ($P_1$) and filial ($F_1$) generations. The 48 h median lethal concentration ($LC_{50}$) of both chemicals was determined. After this, the generations $P_1$ and $F_1$ were exposed for 21 days to three concentrations of glyphosate and Cu mixtures ($1.04$ mg $L^{-1}$ + $2.45$ µg $L^{-1}$, $1.24$ mg $L^{-1}$ + $3.09$ µg $L^{-1}$, and $1.57$ mg $L^{-1}$ + $4.31$ µg $L^{-1}$), starting from neonates. Survival and reproduction were recorded, and macromolecule content and the size of neonates were measured in the progeny. The $LC_{50}$ was $4.22$ mg $L^{-1}$ for Faena® and $13.45$ µg $L^{-1}$ for copper. Exposure to glyphosate and copper reduced fecundity and the number of clutches per female, delayed age at first reproduction, and increased the number of aborted eggs; these effects were more evident in the $F_1$ than in the $P_1$. In both generations, the concentration of carbohydrates and lipids was significantly reduced. The treatment and the brood number influenced the total length of neonates, but the total length did not differ significantly, whereas body length and body width decreased in $F_1$. Glyphosate and copper mixtures significantly increased toxicity in *D. exilis* in the $F_1$ generation, probably because the parents produced impaired progenies. Results warn about transgenerational effects in planktonic species chronically exposed to pollutants.

**Keywords:** zooplankton; herbicides; ecotoxicology; transgenerational effects; water pollution; toxic metals

## 1. Introduction

Freshwater ecosystems are susceptible to chemical pollution, which negatively affects the aquatic biota. In natural environments, organisms are rarely exposed to single pollutants, and the complex chemical interactions among pollutants, producing changes in their toxicity [1], usually are overridden. In an ecological risk assessment, studying the toxic effects of individual pollutants by evaluating reproductive and population effects using reference species is common. However, individual studies are insufficient to determine the toxicity of mixtures and their impact on biota because the action of substances may change due to chemical interactions that modify the endpoints [1,2]. Therefore, in recent years, the assessment of mixtures has become a new way of studying pollutants in the environment and in biological matrices. Nevertheless, it is true that to determine the combined effect of a mixture, it is necessary to previously know the toxicity of each component; therefore, the

individual analysis of components as a traditional approach continues to be essential for the cumulative risk assessment [3].

Glyphosate is a herbicide widely used in agricultural and non-agricultural activities. It is marketed in different chemical formulations worldwide, including in Mexico, where it is sold under different trade names, with Faena® the most popular. Toxicological studies have demonstrated severe damage to aquatic organisms caused by glyphosate-based herbicide exposure, for instance, alteration in reproduction associated with endocrine disruption and oxidative stress [4], inhibition of enzymatic activity [5], and, more recently, cytotoxicity in rat and human cells [6,7]. Furthermore, glyphosate toxic effects have been associated with pesticide mixtures [8,9], but information about their interactions with metals is still scarce [10].

Copper is an oligo-nutrient for all living beings, but, as happens for all the essential metals, when quantities exceed the nutritional requirements, they exert toxic effects. Cu has been used to control fungi and bacteria in agriculture since the 19th century, and nowadays it is routinely used with other pesticides for integrated pest management [11,12]. For example, glyphosate is often applied with the Bordeaux mixture ($CuSO_4$) and may enter waterbodies through runoff and rainwater [13].

Copper is also used in industry and nanotechnology to produce metallic nanomaterials. Due to its characteristics, this metal is incorporated in a wide range of high-consumption products used daily, such as antibacterial agent paints [14], electrochemical sensors [15], lubricants, polymers, plastics, and metallic coatings [16]. This condition favors the release of copper as nanoparticles (NPs) or dissolved metal, increasing their occurrence in soil and water. Exposure to Cu ions is associated with physiologic alterations in aquatic species, such as tissue injury in fish [17], and acute toxicity and CuNP accumulation in microcrustaceans [18].

Because of its molecular characteristics, glyphosate was first patented as a metal chelator, which can form stable complexes with some divalent metal ions, including copper [10]. This chemical property can increase the potential damage that the combination of glyphosate and copper, as commonly used agrochemicals, can produce in aquatic biota.

In the United States, herbicides are nearly ubiquitous in lotic systems, with 90% of the streams located in agricultural, urban, and mixed lands [19]; for this reason, the coexistence of glyphosate and copper is widely found in waterbodies and in soil due to the widespread application of herbicides [13] and the use of Cu in agriculture, including organic horticulture [12].

Constant or periodic exposure to one or more pollutants can modify the population dynamics of aquatic biota in the exposed individuals, across generations, and in multiple generations [20,21]. Hence, the toxicity potential of pollutants could be more evident in the progeny and in future generations of exposed organisms [22]. In many cases, alterations could remain for subsequent generations as a result of the chronic, sub-lethal intoxication of parents. For this reason, it is necessary to determine whether the intoxication of a parental generation can produce measurable consequences in a second generation that was not directly exposed to toxic chemical pollutants. In ecotoxicology, a procedure using *Daphnia magna* as a test organism is suggested to measure the toxic effects in successive generations [23,24]. For other species, Rodriguez-Miguel et al. [21] studied the impact of glyphosate on two generations of the American cladoceran *Daphnia exilis*.

*D. exilis* is a North American freshwater cladoceran distributed from the north to the central part of Mexico. This species thrives from temperate to subtropical latitudes, and because of the similarities with the reference model *D. magna*, it is considered an alternative test organism for ecotoxicological studies in latitudes where *D. magna* is not distributed [25,26].

In daphnids, differences in the content of macromolecules and body size in the progeny can be modified by chemical stress and other environmental factors, such as temperature, photoperiod, type and quantity of food, and the chemistry of the water [21,27,28]; thus,

macromolecule concentration and the size of exposed organisms can be used as biomarkers to assess the effects of toxic pollutants.

In this study, *D. exilis* was chronically exposed to a binary mixture of glyphosate (Faena®) and copper. We aimed to assess toxic effects in the parental generation and to determine if the filial generation can be impaired when the progeny is reared in media free of both pollutants. As endpoints, survival, reproduction, body size in neonates, and the content of the main macromolecules were assessed in the parental and filial generations.

## 2. Materials and Methods

### 2.1. Tested Chemicals

We used Faena® as the commercial presentation of the glyphosate-based herbicide (CAS: 38641-94-0; Monsanto Comercial, S. de R. L. de C. V., Zapopan, Jalisco, Mexico). Faena® contains 363 g $L^{-1}$ of acid equivalent of the potassium salt of N-(phosphonomethyl) glycine (35.6% *w/w* of the active ingredient in the formulation). The other components in Faena® are adjuvants of unknown composition for the consumer (64.4% *w/w*). A stock solution of 500 mg $L^{-1}$ of glyphosate using moderately hard reconstituted water (MHRW) as a medium free of any pollutant was prepared and freshly used in all the experiments and to renew the test solution. The actual concentration of glyphosate in solutions was determined using the Eurofins Abraxis glyphosate ELISA Plate kit, which is an immunoassay for the quantitative and sensitive screening of glyphosate in water samples.

Copper solutions were prepared using the salt $CuSO_4 \cdot 5H_2O$ (J. T. Baker® 98.5% purity, CAS no. 7758-99-8). A stock solution of 20 mg $L^{-1}$ of copper was prepared using MHRW as a dilution medium and refrigerated at 4 °C. The actual concentrations of $Cu^{+2}$ in the experiments were determined following the HACH 10238 method.

### 2.2. Test Organisms

A stock culture of *D. exilis* was started with females of known age according to the conditions stated by Martinez-Jerónimo et al. [26]. Briefly, they were reared in MHRW medium and fed with the green microalgae *Pseudokirchneriella subcapitata* at a concentration of $1 \times 10^6$ cells $mL^{-1}$. Incubation conditions were 25 °C and a photoperiod of 16:8 h (light:dark). Neonates used for the acute and chronic determinations were obtained, starting with the third clutch in the stock cultures.

### 2.3. Acute Toxicity Tests

The median lethal concentration ($LC_{50}$) for glyphosate and copper was determined at 48 h [29]. Glyphosate tested concentrations were 2, 3, 4, 5, 6, and 7 mg $L^{-1}$ based on the glyphosate content in Faena®. The concentrations for copper were 13, 15, 17, 19, and 21 μg $L^{-1}$. MHRW was used as dilution water. Tested concentrations for both toxicants were determined after previously performing range-finding tests; this range enabled an accurate determination of the $LC_{50}$ values.

Tests were prepared with 10 neonates (age < 24 h) distributed in 50 mL beakers containing 35 mL of test volume and three replicates per concentration; MHRW was used for the negative control and dilution water. The temperature for assays was 25 °C and the photoperiod was 16:8 h (light:dark). Immobilization and death of test organisms were recorded at 24 and 48 h to calculate the $LC_{50}$ of both toxicants.

### 2.4. Sub-Chronic Test

Sub-chronic experiments were performed to determine the combined effect of glyphosate (Faena®) and copper in the progenitors ($P_1$ generation) and in the progeny (filial generation $F_1$) of *D. exilis* for 21 days. The combined assays of both toxicants were prepared based on equivalent concentrations of the LC values previously obtained. Three mixtures of both toxicants were assayed by combining the same toxic units of each one in every mixture, i.e., 50% of the $LC_{0.1}$, $LC_1$, or $LC_{10}$ corresponding to each toxicant, estimated through the acute assays.

Neonates of *D. exilis* obtained from stock cultures were used to prepare the experiments for the $P_1$, whereas for the $F_1$ assays, the neonates obtained in each treatment were placed in the same concentrations as for the parents used to obtain the $F_1$ generation. The filial generation test ($F_1$) was prepared using the third clutch obtained from the parental generation ($P_1$). In all cases, one neonate was distributed, for each of 10 replicates per treatment, in 150 mL beakers containing 100 mL of test volume. MHRW was used as negative control and dilution medium in all assays. For both bioassays, females were fed with *Pseudokirchneriella subcapitata* ($8 \times 10^5$ cells mL$^{-1}$). Incubation conditions were 25 °C and a 16:8 h (light:dark) photoperiod. Test solutions were fully renewed every 48 h.

### 2.4.1. Population Parameters

The population responses regarding survival, fecundity (measured as the accumulated progeny), the start of reproduction, and the number of clutches released during the 21-day period were determined through daily observations; these parameters were recorded in both generations ($P_1$ and $F_1$). Additionally, abortions (the non-viable organisms expulsed out of the brood chamber before they completed development) were recorded in both bioassays.

### 2.4.2. Macromolecule Biomarkers

From every clutch obtained in $P_1$ and $F_1$ tests, a sample of 10 neonates was selected, all the water was removed, and 1 mL of potassium-phosphate buffer (100 mM, pH 7.2) was added to preserve the samples at $-20$ °C. For the macromolecule analysis, samples were thawed and processed with a tissue grinder.

Total protein quantification was determined according to Bradford's method [30], adding 1 mL of Bradford´s reagent to 100 µL of the homogenized sample. Processed samples were read at 595 nm, using bovine serum albumin as the standard for the calibration curve.

The content of total carbohydrates was determined according to the method described by Dubois et al. [31]. Briefly, 5% phenol and concentrated sulfuric acid were added to 200 µL of the homogenized sample. Samples were read at 490 nm. Anhydrous serum glucose was used for the calibration curve.

Lipid concentration was determined with the sulfo-phospho-vanillin method [32]. A sample of chloroform:methanol solution (2:1) was added to 500 µL of the homogenized sample; 200 µL of the obtained organic phase was supplemented with concentrated sulfuric acid and evaporated at 90 °C; finally sulfo-phospho-vanillin solution was added to this sample and read at 525 nm, using cholesterol as standard.

### 2.4.3. Neonates' Size

The body size of the progeny was recorded during both bioassays ($P_1$ and $F_1$). From each concentration and for the negative control, 20 neonates were selected randomly and conserved in potassium-phosphate buffer (100 mM, pH 7.2). The total length, body length, and body width were measured using a stereo microscope according to the criteria established in [21] using the software CellsSens® v.1.0.

### *2.5. Data Analysis*

The Probit $LC_{50}$ was determined using the Risk Hazard Assessment Tools, v. 1.0 software. Kaplan–Meier's test was used for the analysis of survival. One way-analysis of variance (ANOVA) was used to compare reproduction parameters, morphological measures, and macromolecule content; normality of data (Shapiro–Wilk test) and homoscedasticity (Bartlett's Test for Homogeneity of Variances) were confirmed before performing the ANOVAs. Multiple pairwise comparisons were conducted according to Fisher's least significant difference (LSD) to determine significant differences ($p < 0.05$) among treatments and control in the $P_1$ and $F_1$ generations. To establish significant differences of treatments with respect to the control, Dunnett's test was performed ($p < 0.05$), specifically for the effects of the treatments on the size of the progeny. The software packages Statistica v. 10.0, GraphPad Prism v. 6.0, and Sigma Plot v. 11.0 were used for the statistical analysis.

## 3. Results

### 3.1. Acute Toxicity of Glyphosate and Copper to Daphnia Exilis

Seven acute toxicity tests for glyphosate were performed, with the coefficients of determination ($R^2$) for the Probit equation values ranging from 0.85 to 0.99. Six acute toxicity bioassays were performed for copper, obtaining $R^2$ values from 0.84 to 0.96. For both toxicants, the fit to the Probit equation was highly significant. The average 48 h $LC_{50}$ was 4.22 mg $L^{-1}$ (3.47–5.43 mg $L^{-1}$ 95% confidence limits), expressed as glyphosate content in the Faena® formulation. The average median lethal concentration ($LC_{50}$) for copper was 13.45 μg $L^{-1}$ (95% limits: 11.48–14.82 μg $L^{-1}$). Additional values of lethal concentration and 95% limits are shown in Table 1.

**Table 1.** Lethal concentration values for *D. exilis* exposed to mixtures of glyphosate and copper. Values in brackets indicate the 95% confidence limits.

| Lethal Concentration (LC) | Glyphosate (mg $L^{-1}$) | Copper (g $L^{-1}$) |
|---|---|---|
| 0.1 | 2.08 (0.76–2.72) | 4.90 (3.03–6.22) |
| 1 | 2.48 (1.02–3.06) | 6.18 (4.10–7.47) |
| 10 | 3.14 (1.63–3.64) | 8.62 (6.28–9.79) |
| 50 | 4.22 (3.47–5.43) | 13.45 (11.48–14.82) |

### 3.2. Survival in Chronic Exposure to Glyphosate and Copper

Figure 1A,B show survival curves of *D. exilis* after 21 days of exposure to glyphosate and copper mixtures in generations $P_1$ and $F_1$. Survival in the control groups of both generations was 100%, and was reduced but not significantly in $P_1$ at all concentrations of glyphosate and copper mixtures. Nevertheless, survival decreased significantly in $F_1$ at all the concentrations of glyphosate and copper (Figure 1B); in the treatment of 1.57 mg $L^{-1}$ glyphosate + 4.31 μg $L^{-1}$ copper, 90% mortality was documented on day 21.

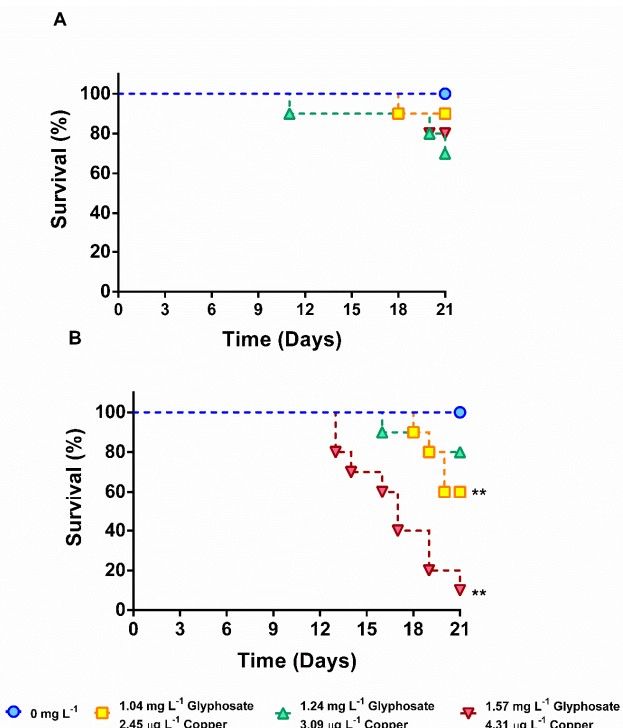

**Figure 1.** Survival of *D. exilis* exposed to sublethal concentrations of glyphosate and copper mixtures (1.04 mg $L^{-1}$ and 2.45 μg $L^{-1}$; 1.24 mg $L^{-1}$ and 3.09 μg $L^{-1}$; 1.57 mg $L^{-1}$, and 4.31 μg $L^{-1}$) for 21 days: (**A**) parental generation ($P_1$), (**B**) filial generation ($F_1$). Differences with respect to the control are marked with asterisks (Kaplan-Meier test, ** $p < 0.001$).

### 3.3. Reproductive Effects in D. exilis Sub-Chronically Exposed to Glyphosate and Copper

Figure 2A shows the total fecundity (measured as the accumulated progeny) determined in $P_1$ and $F_1$ generations of *D. exilis* stressed with glyphosate + copper mixtures for 21 days. The fecundity in the parental generation was not significantly affected in all the treatments; nonetheless, in the filial generation compared with the control, the fecundity was reduced by 44.1, 26.9, and 75.7%, respectively, at the concentrations of glyphosate + copper of 1.04 mg $L^{-1}$ + 2.45 μg $L^{-1}$, 1.24 mg $L^{-1}$ + 3.09 μg $L^{-1}$, and 1.57 mg $L^{-1}$ + 4.31 μg $L^{-1}$.

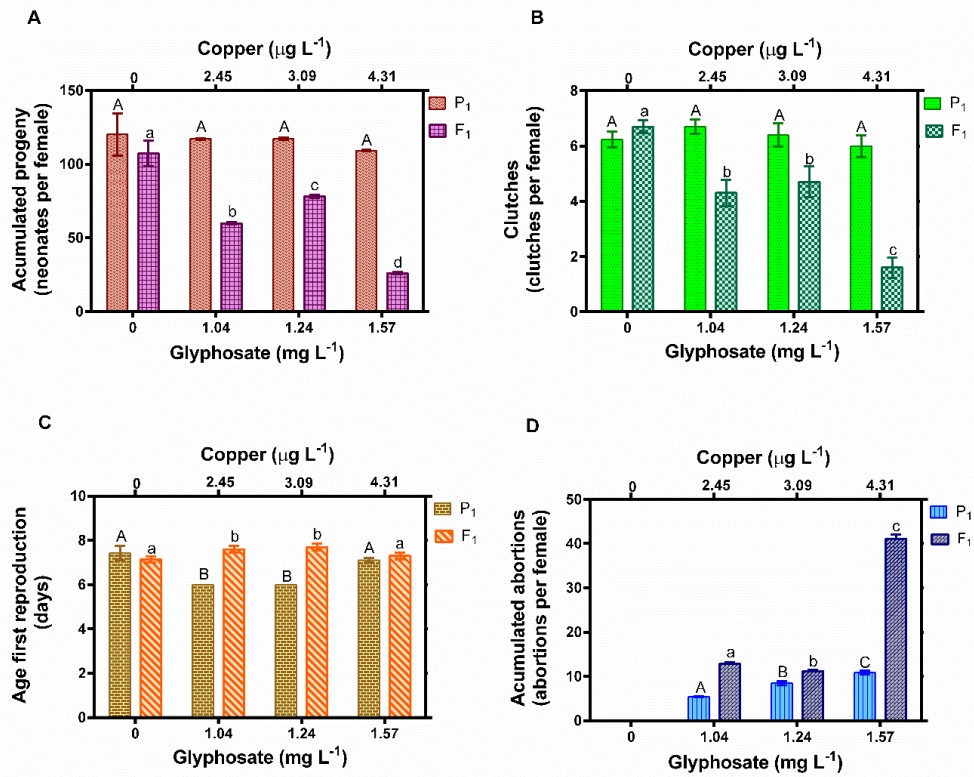

**Figure 2.** Effects in reproductive responses of *D. exilis* exposed to glyphosate and copper mixtures (1.04 mg $L^{-1}$ and 2.45 μg $L^{-1}$; 1.24 mg $L^{-1}$, and 3.09 μg $L^{-1}$; 1.57 mg $L^{-1}$, and 4.31 μg $L^{-1}$) in the parental ($P_1$) and filial ($F_1$) generations: (**A**) accumulated progeny, (**B**) number of clutches, (**C**) age at first reproduction, and (**D**) abortions. Average values ± standard error limits. Different uppercase and lowercase letters are used for significative differences in the $P_1$ and $F_1$ generations, respectively, according to the post hoc LSD test ($p < 0.05$).

The number of clutches per female in $P_1$ and $F_1$ generations of *D. exilis* is shown in Figure 2B. In the parental generation, the number of clutches was not reduced compared to the control ($p < 0.05$) and was higher than that observed in the filial generation. Compared with the control, the number of clutches was significantly reduced in the $F_1$, observing a remarkable reduction in fecundity (76%) in the 1.57 mg $L^{-1}$ glyphosate + 4.31 μg $L^{-1}$ copper mixture.

The age at first reproduction in $P_1$ and $F_1$ is shown in Figure 2C. In the progenitors' generation, females started reproduction at a younger age at the two lowest concentrations of glyphosate + copper treatment than in the $F_1$. In contrast, in the filial generation, the start of reproduction was delayed by 6.6 and 8% with treatments of 1.04 mg $L^{-1}$ glyphosate + 2.45 μg $L^{-1}$ copper and 1.24 mg $L^{-1}$ glyphosate + 3.09 μg $L^{-1}$ copper, respectively, in comparison to the control.

Abortions were recorded in both $P_1$ and $F_1$ generations at all glyphosate + copper treatments but not in the controls (Figure 2D). The number of abortions increased as the concentration of toxicants increased ($p < 0.05$); the highest number of abortions was recorded at 1.57 mg $L^{-1}$ glyphosate + 4.31 μg $L^{-1}$ copper in the $F_1$ generation.

Overall, adverse effects on reproduction increased in the filial generation compared with the outcomes observed in the progenitors.

### 3.4. Macromolecules Content in D. exilis Exposed to Glyphosate and Copper Mixtures

The contents of proteins, lipids, and carbohydrates measured in the progeny in the $P_1$ and $F_1$ generations of *D. exilis* after exposure to glyphosate and copper mixtures are shown in Figure 3. Protein content was affected only in the filial generation ($F_1$) in the mixture of 1.24 mg $L^{-1}$ of glyphosate + 3.09 µg $L^{-1}$ copper (Figure 3A). In the parental and filial generations, lipids were significantly reduced in the two lowest combinations of glyphosate and copper, but the carbohydrates decreased at all concentrations with respect to the control. In $P_1$, the reduction in carbohydrates was 68, 95, and 94% for the glyphosate and copper mixtures (1.04 mg $L^{-1}$ + 2.45 µg $L^{-1}$, 1.24 mg $L^{-1}$ + 3.09 µg $L^{-1}$, and 1.57 mg $L^{-1}$ + 4.31 µg $L^{-1}$, respectively). The content of carbohydrates was higher in $F_1$ than in $P_1$ at the two highest concentrations of toxicants.

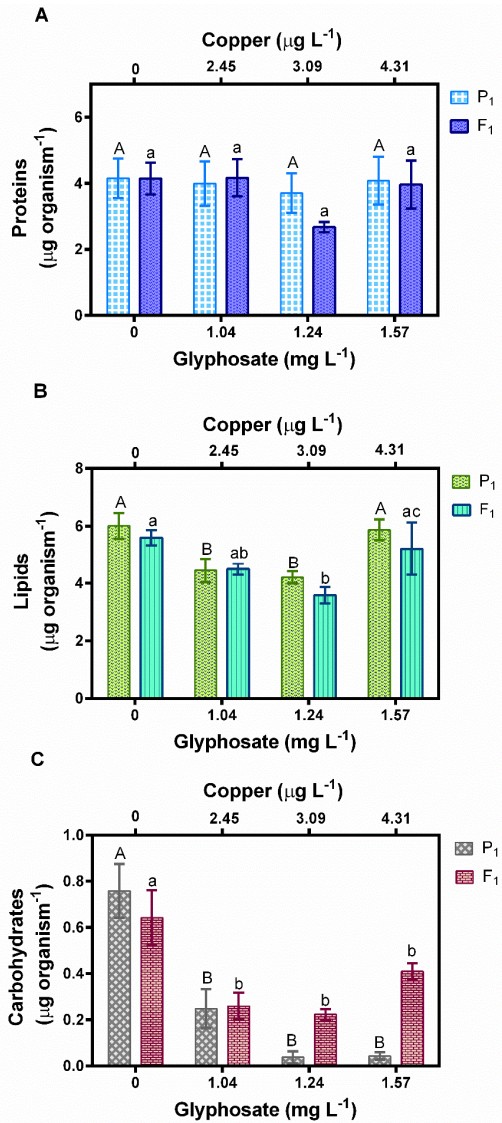

**Figure 3.** Macromolecule concentration determined in neonates of *D. exilis* exposed for 21 days to glyphosate and copper mixtures (1.04 mg $L^{-1}$ and 2.45 µg $L^{-1}$; 1.24 mg $L^{-1}$, and 3.09 µg $L^{-1}$; 1.57 mg $L^{-1}$, and 4.31 µg $L^{-1}$) in the parental ($P_1$) and filial ($F_1$) generations: (**A**) proteins, (**B**) lipids, and (**C**) carbohydrates. Average values ± standard error limits. Different uppercase and lowercase letters are used for significative differences in the $P_1$ and $F_1$ generations, respectively, according to the post hoc LSD test ($p < 0.05$).

### 3.5. Effects of Glyphosate and Copper Mixtures on the D. exilis Neonate Size

The body measures of neonates produced in consecutive clutches during the P1 and F1 generations are presented in Figure 4A–C; missing information is for clutches without enough neonates to be measured. TL and BL of neonates in the parental generation displayed reduced variation in the controls. In the controls of $F_1$, a similar situation was observed, except for clutch number four, in which neonates were significantly larger. According to Dunnett's test, TL in both generations ($P_1$ and $F_1$) did not reveal significant differences with respect to their control groups, except for some clutches (Figure 4A).

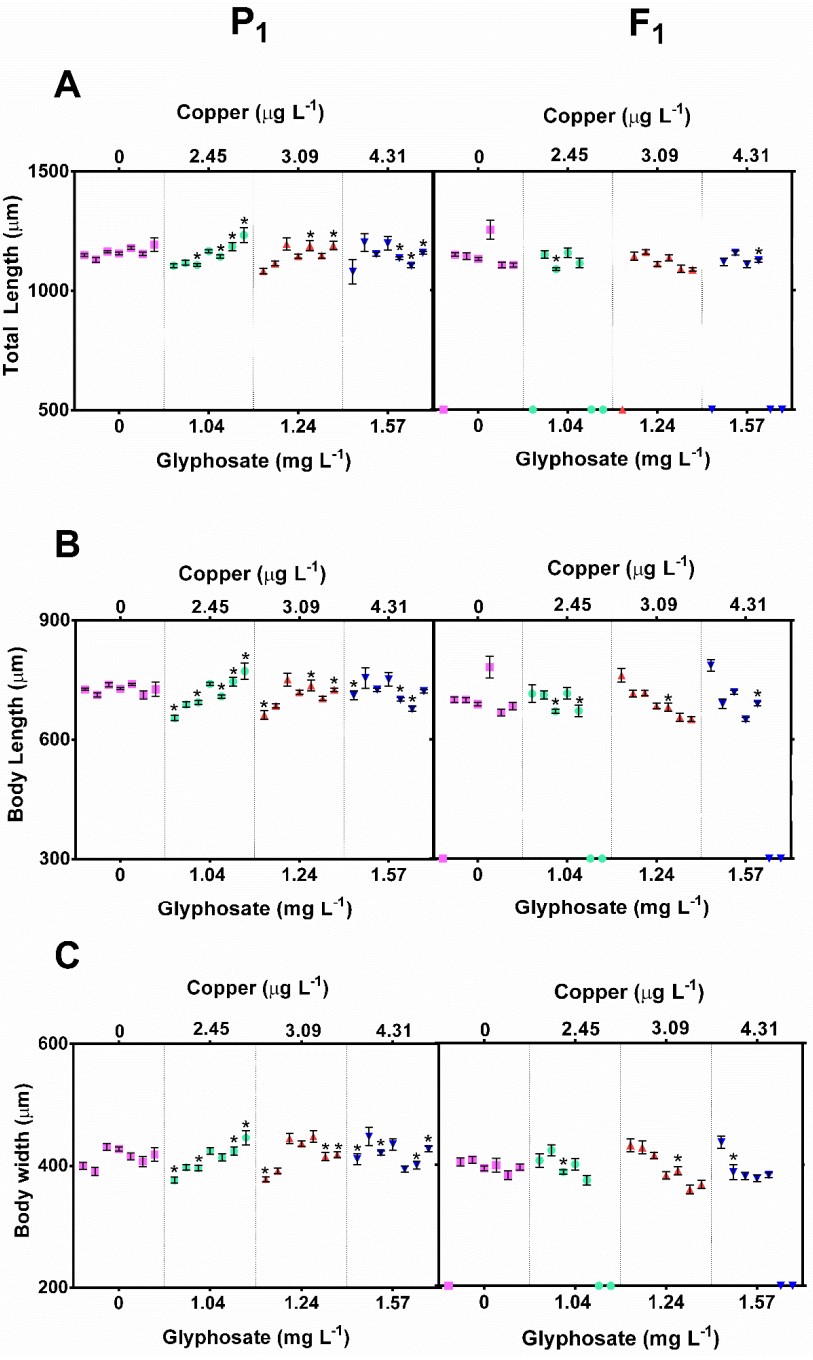

**Figure 4.** Body size of the neonates produced in progenitors exposed to glyphosate and copper mixtures in generations $P_1$ and $F_1$ (left and right, respectively): (**A**) total length (TL), (**B**) body length (BL), and (**C**) body width (BW). Significant differences comparing with the controls are represented by asterisks (Dunnett's test; $p < 0.05$). Mean values $\pm$ standard error limits.

The body length of neonates in the parental generation ($P_1$) was larger in some clutches in the three tested treatments; however, this effect was not observed in the filial generation. In $F_1$, the BL decreased as the number of the clutch event increased. The BW (Figure 4C) varied similarly to BL (Figure 4B).

## 4. Discussion

Glyphosate and copper are common agricultural products that can be found as toxic pollutants in aquatic environments. More than 130 commercial formulations of the herbicide glyphosate are currently commercialized for application in different crops in agriculture [33]. The present study documented significant adverse effects of glyphosate and copper on the reproduction and survival of *Daphnia exilis*. For glyphosate, with the formulation known as Faena®, similar results to those observed here were reported by Rodríguez-Miguel et al. [21] in *D. exilis* ($LC_{50}$: 4.22 mg $L^{-1}$), and Domínguez-Cortinas et al. [5] in *D. magna* ($LC_{50}$: 4.1 mg $L^{-1}$). Lares et al. [34] observed that Faena® was 12 times more toxic to *D. exilis* than to *Alona guttata*. Reno et al. [35,36] determined the 48 h $LC_{50}$ for *D. magna* exposed to the four glyphosate-based formulations, Escoba®, Roundup®, Ultramax Panzer Gold®, and Sulfosato Touchdown®; the obtained values were 29.48, 11.68, 2.12, and 1.62 mg $L^{-1}$, respectively. The differences in the acute toxicity of glyphosate (as Faena®) in the present work and the other commercial formulations could be attributed to the active ingredients and the type and concentration of surfactants and other adjuvants used in the different formulations. In this respect, Lares et al. [34] reported that the toxicity of commercial formulations in *D. magna* and *C. dubia* exceeded the toxicity of active ingredients by 77-fold. Tsui and Chu [37] reported that, in Roundup®, the surfactant POEA (polyoxyethylene amine) was more toxic to *C. dubia* than the formulation and the active ingredient. Nevertheless, most of the toxicity information on glyphosate and its formulations refers to short-term, acute toxicity measurements.

Cladocerans have been proved to be the most sensitive of zooplankters to the toxicity of copper [38]. For copper, the $LC_{50}$ determined in *D. exilis* was 13.45 µg $L^{-1}$. Fan et al. [39] reported that, in *D. magna,* the 24 h $LC_{50}$ for this metal was 45 µg $L^{-1}$. The 48 h $LC_{50}$ value for *D. carinata* exposed to Cu was 14.79 µg $L^{-1}$ [40] and 37.3 µg $L^{-1}$ [41]. Banks et al. [42] determined that the 48 h $LC_{50}$ value for copper in *Ceriodaphnia dubia* was 12.0 µg $L^{-1}$ (8–18.8 µg $L^{-1}$). Differences in toxicity among cladocerans could be due to species-specific differences related to different capabilities to accumulate, metabolize, and eliminate the toxicants [34].

The mixture of the herbicide Faena® and copper produced effects in the content of macromolecules in both generations of *D. exilis* that are related to physiological alterations; this impairment finally affected survival and reproduction. We observed that the effect of this mixture at all the tested concentrations increased in the filial generation with respect to the parental generation. Other studies have reported the single effect of glyphosate and copper on the survival of cladocerans. Rodriguez-Miguel et al. [21] described that the survival in the parental generation decreased by 50% when exposed to 2.49 mg $L^{-1}$ glyphosate (Faena®). Survival in *D. magna* decreased significantly at 4.05 mg $L^{-1}$ of Roundup® [43]. Furthermore, for *D. magna*, Fan et al. [39] documented that the survival rate decreased from 100% to 0% as the Cu concentration increased from 10 to 100 µg $L^{-1}$. Koivisto et al. [44] reported that cladoceran sensitivity to copper is species-specific and that mortality increased at low food concentrations (2000 to 20,000 *Scenedesmus* cells $mL^{-1}$, depending on the species), indicating that the toxicity of this metal also depends on feeding conditions.

However, the effect of binary mixtures as tested here has not been assayed often in cladocerans; for example, Dong et al. [40] reported that in *D. carinata* the mortality increased 6–7% on the first 5 days of exposure with 3 µg $L^{-1}$ of copper combined with microplastics. Mahar and Watzin [45] found that the binary mixture of Cu and Zn affected the survival and reproduction of *C. dubia*. Banks et al. [42] found that the survival of *C. dubia* was affected by the concentrations of copper and diazinon in the mixture. In

the present study, survival in the filial generation ($F_1$) decreased 20, 40, and 90% with glyphosate and copper mixtures (1.04 mg L$^{-1}$ + 2.45 µg L$^{-1}$, 1.24 mg L$^{-1}$ + 3.09 µg L$^{-1}$, and 1.57 mg L$^{-1}$ + 4.31 µg L$^{-1}$, respectively). The increase in sensitivity observed in the $F_1$ demonstrated transgenerational effects; this increase in the toxic response of the progeny could also mean risks for subsequent generations that could affect the persistence of the population in natural conditions [21].

The toxic effects of Faena® and copper mixtures found in the present study demonstrated reproductive impairment in *D. exilis*, evidenced by reduced fecundity, decreased number of clutches per female, delayed age at first reproduction, and increased number of aborted eggs; all these effects were mainly observed in the $F_1$. Similar results have been described for copper and glyphosate tested separately. For example, refs. [46–48] reported that, in *D. magna*, *D. pulex-pulicaria*, and *D. longispina,* reproduction was delayed when exposed to Cu. In *Bosmina longirostris* and *Chidorus sphaericus*, the total number of offspring was significantly reduced when cladocerans were exposed to Cu [44].

We observed adverse effects on the start of reproduction, and on the number and size of released clutches, mainly in $F_1$ of *D. exilis*. Concerning the toxicity of glyphosate on cladocerans, when *D. magna* was exposed to 2.92, 4.38, 6.57, 9.86, and 14.86 mg glyphosate L$^{-1}$, age at first reproduction was delayed with all concentrations, and at 14.86 mg L$^{-1}$ this delay was up to two days [49]. In *D. magna* exposed to Roundup®, the fecundity and the number of neonates per brood in the maternal generation (I) and in generations II–IV were reduced significantly compared with the control [50]. Villarroel et al. [51] observed that offspring in *D. magna* exposed to tetradifon decreased in $F_1$. The number of broods of *Daphnia carinata* exposed to copper and microplastics (3 µg L$^{-1}$ and 0.25 mg L$^{-1}$, respectively) was reduced by 9% [40]. Delay in the sexual maturity of organisms has been observed in daphnids under chemical stress [34,52].

Reduction in clutch size does not seem to be a typical response to sub-lethal toxic metal stress, and the lifetime reduction in offspring production is generally a consequence of increased mortality [44,53,54]. Toxic effects of Faena® and copper in *D. exilis* might lead to impaired vital functions, leading to increased chronic mortality in adults and reduced fecundity. Under stress conditions, organisms face a tradeoff to endure the harmful effects of toxic chemical stressors, maintaining some functions at a basic level and devoting energy to ameliorate the effects of stressors, resulting in a reduction in the energy available for reproduction and for the reserves and growth in the progeny [55].

In our study, the number of abortions increased as the concentration of glyphosate and copper in the mixture increased, both in the parental and filial generations. Similarly, Rodriguez-Miguel et al. [21] documented aborted eggs in *D. exilis* at all the tested glyphosate (Faena®) concentrations, with the highest number of abortions for 3.15 mg L$^{-1}$ glyphosate. Cuhra et al. [43] and Gill et al. [56] reported that 1.35 mg L$^{-1}$ of Roundup® delayed the start of reproduction and produced 100% abortions in *D. magna*.

Glyphosate could act as a chelating agent to form stable complexes with copper [57]. These complexes are stable in water and can modify the toxicity of glyphosate and copper as single toxicants. However, scarce information is available about the aquatic toxicity of these chemical aggregations [10] and the alteration they produce in biochemical processes, such as macromolecules' synthesis. In this study, we demonstrate that the most affected macromolecule was the concentration of carbohydrates, followed by the contents of lipids and proteins, in both generations. Rodriguez-Miguel et al. [21] mentioned that proteins and lipids decreased at all sub-lethal Faena® concentrations assessed. Similarly, Villarroel [58] observed a decrement in the three main macromolecules in neonates of *D. magna* produced by exposure to the herbicide propanil. The reduction in lipids and proteins under conditions of toxic stress could also be due to different physiological mechanisms that affect the formation of lipoproteins, which are used to repair the damage in cells and tissues [59]. In our study, a change in macromolecule content could have resulted from toxic effects produced by glyphosate and copper mixtures.

Chemical toxic stress triggers compensatory processes that require energy; this response forces a redistribution of the available energy resources, reducing the energy destined for reproduction and growth [60]. Thus, the exposure of cladocerans to chemical stressors will produce negative effects at metabolic levels due to decrements in energy reserves and damage to growth and reproduction [61]. In *D. exilis*, reduction in the levels of carbohydrates and lipids as the primary energy sources drive organisms to use their available energy to stay alive, leading to less reproduction and growth; in this way, this cladoceran manages to counteract the toxic effects that the glyphosate and copper mixture is causing.

In polyembryonic organisms, such as the cladocerans *Simocephalus* and *Daphnia*, during the life cycle the size of neonates is typically variable, with the first reproductions being of smaller neonates; the number of neonates in each clutch is also variable [62]. Reno et al. [63] mentioned that when organisms are exposed to pollutants, cladocerans invest energy to survive at the cost of other biological functions such as reproduction or growth. During Cu exposure, reduced weight and body length were observed in *Daphnia*, resulting in an impaired growth rate [64]. In chronic exposure of *D. magna* to glyphosate and Roundup®, Cuhra et al. [43] documented a significant reduction in juvenile size even at the lowest concentrations they evaluated. Our results demonstrated that, in the $F_1$ generation, the size of *D. exilis* neonates was significantly smaller than that of the neonates produced in $P_1$. Papchenkova et al. [50] observed similar results for morphological growth, reporting that the growth in juveniles of *D. magna* exposed to glyphosate was significantly lower compared with the controls and in successive generations. Rodriguez-Miguel et al. [21] found that exposure of *D. exilis* to Faena® provoked smaller adults in $F_1$ than in $P_1$. The reduced size of cladocerans can result from the chemical stress produced by active ingredients and, in commercial formulations (v. gr. Faena®), other ingredients such as surfactants, adjuvants, and solubilizers also contribute to this effect.

Cladocerans, one of the three main zooplankton groups in freshwater bodies, are exposed to chemical stressors in polluted environments. In particular, the extended use of glyphosate in all the commercial formulations currently available adds to the use of other agrochemicals, such as copper salts, which can be found together in agricultural effluents and runoffs that can reach freshwater environments. The multi-generational and transgenerational effects of chemical pollutants are of great concern now because wild populations could be affected, threatening their persistence and modifying the structure of aquatic communities. Our results demonstrate that the combination of one glyphosate formulation with copper produced significant adverse effects in the filial generation of *D. exilis*. This information confirms impairments of different magnitudes and levels, ranging from biochemical to reproductive affectations caused by the tested pollutants.

## 5. Conclusions

The sensitivity of *D. exilis* to glyphosate and copper mixtures was evaluated at very low concentrations (1.04–1.57 mg $L^{-1}$ for glyphosate and 2.45–4.31 µg $L^{-1}$ for copper), which can be found in aquatic environments. These chemicals affected survival and reproduction in both generations ($P_1$ and $F_1$) and induced abortions. The biochemical changes in *D. exilis* during exposure to both toxicants can be associated with the effects observed in reproduction and growth. The glyphosate and copper mixtures caused increased toxic effects in the $F_1$ generation of D. *exilis*, probably because, in the parents, the damages were counter-balanced by using energy stores to endure the toxic effects. However, the sensitivity in the progeny was affected, probably due to a reduction in lipid reserves or a modification in the balance of the main macromolecules, caused by the intoxication of the progenitors. Although the effect of each toxicant was not assessed separately in the sub-chronic tests, and it was not possible to determine if chemical interactions between glyphosate and copper happened, toxic transgenerational effects were documented in *D. exilis*. However, it also should be considered that because the ingredients and concentrations in the commercial formula of glyphosate are unknown, the effect of the varying concentrations of adjuvants

and other constituents in the dilutions of Faena® could also modify the toxicity of the active ingredient. Nevertheless, it is relevant for the environment that the biological effects of the mixture of toxicants could be determined despite the possible chemical interactions, as the test organisms were able to integrate the effects of all the toxicants in the biological responses they exhibited.

**Author Contributions:** Conceptualization, M.H.-Z. and F.M.-J.; Formal analysis, M.H.-Z. and F.M.-J.; Funding acquisition, F.M.-J.; Investigation, M.H.-Z., A.R.-M., L.M.-J. and F.M.-J.; Methodology, M.H.-Z., A.R.-M., L.M.-J. and F.M.-J.; Writing—original draft, M.H.-Z., A.R.-M. and F.M.-J.; Writing—review and editing, F.M.-J. All authors have read and agreed to the published version of the manuscript.

**Funding:** This research received no external funding.

**Data Availability Statement:** All the data supporting the results presented here are available under request.

**Acknowledgments:** This study was supported by the Instituto Politécnico Nacional (SIP), Secretaría de Investigación y Posgrado. F. Martínez-Jerónimo thanks the COFAA-IPN and the EDI-IPN for grant support. Thanks to three anonymous reviewers because their comments and suggestions improved this paper.

**Conflicts of Interest:** The authors declare that they have no known competing financial interests or personal relationships that could have appeared to influence the work reported in this paper.

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
