# Peer review of "Combined Toxicity of Glyphosate (Faena®) and Copper to the American Cladoceran Daphnia exilis—A Two-Generation Analysis"

_water, doi:10.3390/w15112018_

Round 1
Reviewer 1 Report
The present study was designed to investigate the combined effects of glyphosate and copper on the Daphnia exilis in two generations. Overall the study is well designed and drafted and provides some interesting information. However, some details need to be modified carefully before publication.
The reason for selecting the exposure concentration of these chemicals should be addressed in the manuscript. And If you think these concentrations are environmentally relevant, as you mention in line 457, you need to bring a range of detected concentrations in the aquatic systems.
To run ANOVA, your data needs to meet the assumptions of parametric tests. Please add this information to the data analysis section.
The toxicity of these chemicals separately and in combination has been reported in other species; I recommend that the authors highlight the innovation of the article.
In the discussion section, you should have first written the significance of your results and then brought different examples. For example, in lines 316-332, you suggested "similar results" were observed by other researchers without any information regarding your results. Again in lines 354-359, you should first clarify what your results show and then compare them with other studies.
Line 362: "The increase in sensitivity observed in F1 demonstrated transgenerational effects". This sentence does not make sense as you also exposed F1 to the chemicals. For this to be considered a transgenerational effect, individuals from F2 would need to be assessed.
Lines 354-365: it is confusing why you compare the results of other studies with yours when they used different chemicals with different concentrations and even different species.
Line 345 is vague" This effect produced that fecundity was reduced in the F1."
What do you mean with low concentrations, Line 352, 456?
Minor editing is required, particularly in the discussion section.
Reviewer 2 Report
Although the methodology basically sounds, the experimental design used for the chronic assay is criticizable, as mentioned below. Statistical methods and expression of results should be also revised.
L.112-113. I think that a “media free of any pollutant” would be a better statement
L.129. Do the authors mean the “actual concentration of Cu++?
L.131. A bibliographic reference to the mentioned method should be provided
L. 141. Which was the rationale for using these concentration series? To estimate lethal values, the experimental concentrations commonly follow a logarithmic series
Table 1 is in principle out of place, since it is based on the results of the acute assays aimed at determining LC50 values. Moreover, the caption and content of the first two columns are confusing or wrong. Therefore, Table 1 could be deleted, modifying the text in M&M; I suggest something like “Three mixtures of both toxicants were assayed by combining in every mixture the same toxic units of each one, i.e., the 50% of the LC0.1, LC1 or LC10 corresponding to each toxicant, estimated by means of the acute assays”
However, since in the sub-chronic test the authors did not assay each toxicant separately, at the same LC values of the mixtures, no information about the possible interaction between glyphosate and cooper (addition, synergism, or antagonism) could be obtained. I think this is a weak point of the study. I recommended the authors explore more deeply the literature about the join toxicity of toxicants.
Statistical analysis (L.212-215): it is unclear which post-hoc comparison test was used to compare control with every mixture: Dunnet or LSD? Besides, a two-way ANOVA sounds more suitable to analyze the experimental data, taking as factors: treatment (control and mixtures), and generation (P and F1). In this context, interaction comparisons could have been made, i.e., control versus each mixture, for every generation; and also, P versus F1 for every treatment (control or mixture). This latter comparison is not reported in the current results.
Tables 2. The 0.1, 1, and 10 LC values for copper are far below the minimal concentration used in the acute assay. In addition, the estimation of LC0.1 and LC1 is not consistent with the number of animals used. This would imply an extrapolation of the regression line in the probit analysis. A table stating R values and other parameters of the analysis should be necessary to validate these results.
Labels in most Figures are unreadable. Please correct. Besides, I think that only one X-axis should be drawn, labeling the total concentration presented in the mixtures
Figure 4: Which control/s are referring to the asterisks? Why an overall mean value is not reported for every treatment? Why in this case the Dunnet test was used, and the LSD for the other variables (Figs. 2 and 3)? The statistical analysis should be consistent
Discussion: since the authors did not assay each toxicant separately, they are unable to evaluate the relative contribution of each toxicant to the results observed in the chronic assay, and neither the kind of interaction between them. This is a serious limitation of the study
Although the English style is understandable, the text would benefit from a careful revision
Reviewer 3 Report
none
The manuscript needs editing to address grammar
Round 2
Reviewer 2 Report
The manuscript has been improved. However, there are some issues that in my opinion should need further revision, to know:
- Labels of most Figures are still difficult to read, especially Figs. 1 and 4. The authors could try another graphing program that ensures better art quality, also in accordance with the journal's requirements.
- The statistical analysis remains inconsistent. The choice of a multiple comparison method depends on which kind of comparisons the researchers are interested in. Therefore, the analysis and interpretation should be made in a particular context for a given data set. For example, if only the comparison of each treatment against the control is of interest, Dunnett’s test could be employed. On the other hand, if all possible comparisons by pairs were relevant, including the comparison of each treatment with control, a different comparison method should be chosen, such as LSD. But applying two different comparison methods for reporting the analysis of the same data set is not correct, because the interpretation should be made in a unique context.
That said, the multiple comparisons shown in Figs. 2 and 3 seem to have been made by only using the LSD method. I think this is OK. However, Dunnett’s test was used for data shown in Fig. 4. Why should a morphological variable, such as body length, be analyzed in a different context than a biochemical one, such as the protein content? Does it have to do with the fact the results of different clutches are shown, and in this context, the authors are only interested in highlighting the comparison of each treatment with the control? If so, this reason (or eventually any other) should be clearly stated in Materials and Methods (L.212-214), to justify the election of a different comparison method.
Finally, the epigraph of Fig mentions a statistical method (Kaplan-Meier) to compare the percentages of survival. This method should be referred to in Material and Methods.
No comments
